**communications** engineering

# A physically motivated voltage hysteresis model for lithium-ion batteries using a probability distributed equivalent circuit
Leonard Jahn [1,2], Patrick Mößle [1,2], Fridolin Röder [1,2] ✉ & Michael A. Danzer [1,2] ✉

The open circuit voltage hysteresis of lithium-ion batteries is a phenomenon that, despite intensive research, is still not fully understood. However, it must be taken into account for accurate state-of-charge estimation in battery management systems. Mechanistic models of the open circuit voltage hysteresis previously published are not suitable for deployment in a battery management system. Phenomenological models on the other hand can only superficially represent the processes taking place. To address this limitation, we propose a probability distributed equivalent circuit model motivated by the physical insights into hysteresis. The model incorporates hysteresis effects that are often disregarded for state estimation, while keeping the computational cost low. Although the parameterization is more demanding, the model has the advantage of providing insight into the internal state of the battery and intrinsically incorporating the effect of path-dependent rate capability.

Lithium-ion batteries (LIBs) are used in portable devices, stationary battery energy storage systems, and battery electric vehicles. Accurate knowledge of the current state of charge is essential for safe and efficient operation[1–3]. In the battery management system the state-of-charge (SOC) is often determined by Coulomb counting, i.e., by measuring and integrating the current at the terminals of the battery. This method has the disadvantage that measurement errors cumulate over the time the system is used[1]. To correct these errors, the cell voltage can be used, as the open circuit voltage (OCV) depends on the SOC. Unfortunately, the OCV can also depend on the charge and discharge history; this particular type of path dependency phenomenon is known as voltage hysteresis[4]. For a correct SOC estimation it is therefore necessary to consider the voltage hysteresis.

Voltage hysteresis is a short-term path-dependent phenomenon[4], as the OCV depends on the short-term history of the applied current. However, even after very long relaxation, i.e., an operation with no current applied, voltage hysteresis does not vanish[5–7]. Voltage hysteresis is strongly dependent on the active material. For example, graphite with ~10 mV[8], lithium iron phosphate (LFP) with up to 20 mV[5] and silicon (Si)[9] with more than 200 mV are known to have pronounced voltage hysteresis, while lithium-titanate (LTO) shows negligible hysteresis. Moreover, the underlying mechanism may be different, e.g., first-order phase transition for LFP[10,11] or charge direction dependent reaction pathways for Si[12,13]. Hysteresis can also be visualized, for example, by the color of graphite particles, which has been shown to be dependent on current direction[14]. In addition, voltage hysteresis can be related to other path-dependent phenomena such as the path dependency of cell expansion[15] as well as rate and power capability[16–18]. Therefore, limiting the assessment to the description of voltage hysteresis may not be sufficient.

To account for the hysteresis of the OCV and thus improve the accuracy of the SOC estimation, a mathematical description of the OCV as a function of the current direction and the stored amount of charge relative to the cell's capacity is required. Many approaches have been described in the literature, some of which are generally applicable and some of which are specifically designed for certain cell chemistries. Plett discusses state estimation approaches in detail in his work[19–21]. He extends an equivalent circuit model (ECM) for describing the dynamic hysteresis behavior of LIBs with a zero-state response model, where a switch between charge and discharge is considered by adding or subtracting the voltage hysteresis to the OCV equation, respectively. Plett extends this approach by adding a single hysteresis state[20]. In this model, the voltage hysteresis is accounted for by an additional differential equation. Motivated by hysteresis models in magnetic fields, an adaptation of the Preisach model for LFP is proposed by Tjandra and Jossen[22] and Baronti et al.[23]. Some publications developed mathematical functions to model the voltage hysteresis of LFP, especially the transition paths between charge and discharge curve[24,25]. Other groups have also used a long short-term memory neural network to learn models that can describe voltage hysteresis of LFP[26,27]. For materials with very pronounced hysteresis such as Si, more specialized hysteresis models were used. A Prandtl-Ishlinskii model, based on Preisach modeling, was proposed by Chayratsami et al.[28]. Baker et al.[29] developed mathematical formulae to describe the hysteresis of Si, that were later applied by Graells et al.[30] for the voltage prediction of Si/graphite half-cells. Both models have additional degrees of

¹Chair of Electrical Energy Systems (EES), University of Bayreuth, Universitätsstraße 30, 95447 Bayreuth, Germany. ²Bavarian Center for Battery Technology, University of Bayreuth, Universitätsstraße 30, 95447 Bayreuth, Germany. ✉e-mail: fridolin.roeder@uni-bayreuth.de; danzer@uni-bayreuth.de

freedom to design the transition path of the voltage hysteresis, describing the large and lithiation-dependent hysteresis path of Si in more detail.

The models discussed so far are useful for the application in battery management systems (BMS), but they are only phenomenological descriptions of the experimentally measured behavior. Mechanistic models are an alternative, but are often associated with high computational cost, which makes their application in BMS difficult, though not impossible[31]. Farkhondeh et al.[32] proposed a mesoscopic model consisting of multiple units with an underlying non-monotonous equilibrium potential and a resistance distribution. A similar approach was followed by Kondo et al.[33] using multi-particle model with a particle size distribution. Jiang[13] simulated crystallization and reaction pathways for Si to predict voltage hysteresis. These models provide a deeper insight into the internal state of the cell, e.g., the particle-by-particle phase change. In addition, these models have been shown to predict not only the voltage hysteresis, but also the path dependency of the rate and power capability. As prediction of rate and power capability is another important task for the BMS, these mechanistic models provide an important added value. Furthermore, the estimation of the remaining useful lifetime of the cell is another important topic for battery management, where hysteresis of OCV and impedance are of relevance[34,35]. However, a significant reduction in the computational cost is required for application. Additionally, mechanistic models are usually limited to certain electrode materials.

In this article, we propose a modeling approach based on a classical equivalent circuit of electrical networks, which, due to its mathematical structure, is applicable in BMS. The model is characterized by distributing the cell capacity into parallel strings to account for string-by-string charging, analogous to the particle-by-particle charging in frequently reported mechanistic models. Thus, this model provides detailed information on the internal state of the voltage hysteresis and the path dependency of the rate and power capability. The described structure is based the mechanism of materials that undergo a first-order phase transition. Therefore, the model is not suitable for other materials that utilize different lithium-storage mechanisms, such as silicon. To the best of the authors' knowledge, this is the first time that the physical principles of hysteresis in materials such as LFP have been modeled using an ECM. The model reaches real-time capability and enables an easier integration of the frequently overlooked impact of rate capability hysteresis into BMS. In this article, the operating principle of this novel hysteresis model is explained, the effect of parameter variations is investigated and the model is applied to simulate the voltage curve of a commercial cell. In addition, the results are compared with well-established hysteresis models based on an ECM from the literature.

## Results and discussion

The proposed hysteresis model is suitable for battery materials where the hysteresis is caused by a first-order phase transition. In this case, the Gibbs free energy has two minima, so there are two stable phases in the two-phase region, a lithium-poor phase ($\alpha$) and a lithium-rich phase ($\beta$)[36]. The electrode potential is proportional to the change in Gibbs free energy during insertion/extraction of lithium ions, so the open circuit potential (OCP) without phase separation is a non-monotonic function of the degree of lithiation[10]. The OCP has a local maximum at a slightly higher lithiation state than the $\alpha$-phase and a local minimum at a slightly lower degree of lithiation than the $\beta$-phase. In the case of phase separation, the $\alpha$ and $\beta$ phases can coexist in equilibrium because they have the same potential. In a multi-particle system this can lead to a particle-by-particle phase change during lithiation/delithiation. During lithiation, all particles are filled with lithium until they reach the local maximum of the non-monotonic potential curve. As soon as the first particle crosses this maximum, it is lithiated until it reaches a stable state in the Li-rich phase with the same potential. This leads to sequential lithiation of the particles. The potential of the multi-particle system with phase separation is the horizontal line at the local maximum of the non-monotonic potential curve. During delithiation, the potential of the multi-particle system with phase separation is again a horizontal line, now starting from the local minimum of the non-monotonic potential curve[5].

The difference of the maximum and minimum is observed as voltage hysteresis. The sequential order of particle lithiation/delithiation depends on the overpotentials of the individual particles. For example, poorly connected particles have a higher ohmic resistance and will therefore be lithiated/delithiated later.

The hysteresis theory described above is now transferred into a physically motivated ECM. Therefore, a non-monotonic OCP, i.e., $OCP_n$ and a distribution of cell capacity in multiple parallel strings are considered, which allows to model the voltage hysteresis of phase-separating materials such as LFP. In addition, a distribution of electrical resistances within the multiple strings is used. The detailed description of the model, its mathematical representation, and the solving algorithm are given in the Methods section. The model is used to simulate three basic tests, (1) a pseudo OCV test, (2) a pseudo OCV test with partial cycles and (3) a path dependency test for rate capability. The tests are also described in detail in the Methods section. The model is parameterized with experimental data and simulation errors are analyzed and compared with another hysteresis models from the literature.

### Characteristics of the probability distributed ECM

Figure 1 shows simulation results for the pseudo OCV test. An ECM with only $N = 10$ parallel strings and normally distributed resistance values is used to illustrate the model behavior. The underlying non-monotonous curve is shown in Fig. 1a. It has a maximum at a SOC of 0.4 and a minimum at a SOC of 0.7. The simulated voltage is shown as the green line in Fig. 1a. Voltage versus time and applied C-rate is shown in Fig. 1b. It can be seen that the voltage of the pseudo OCV test differs from the considered $OCP_n$. For the charging phase, the voltage remains almost constant after the maximum of the $OCP_n$ at 40% SOC has been exceeded. During the subsequent discharge, the voltage shows a similar behavior, this time though after passing the local minimum of the $OCP_n$ at 70% SOC. The model can therefore represent the voltage hysteresis. However, it can also be seen that there are significant ripples in the voltage as it moves horizontally.

For a deeper insight into the characteristics of the model, the course of the $OCP_n$ and the SOC within the parallel strings, called $SOC_n$, are shown over time in Fig. 1c, d, respectively. The number of parallel strings is indicated in the color bar, where larger string numbers $n$, correspond to a higher assigned resistance value $r_n$. In Fig. 1d, initially the $SOC_n$ almost simultaneously increases for all parallel strings. Slight differences can be seen, as strings with a smaller assigned resistance have a slightly higher $SOC_n$. Therefore, the parallel string with the smallest resistance, i.e., $n = 1$, reaches the $SOC_n$ with the local maximum of the potential curve first. As soon as a string exceeds the $SOC_n$ of 0.4 the $SOC_n$ quickly rises to 0.8. Thus, a sequential charging of the parallel strings, i.e., string-by-string charging, is observed. The sequential order is determined by the assigned resistance, while strings with lower resistance are charged earlier. Figure 1c shows, that as soon as the first string crosses the local maximum, the $OCP_n$ in this string is decreased, leading to a fast charging of a single string. Furthermore, in Fig. 1d it becomes apparent that during these phases, the $SOC_n$ of all the other strings decreases slightly, indicating that the charge is mainly redistributed and supplied by the other parallel strings. This behavior also explains the observed ripples in the simulated voltages as it is associated with significant changes of $OCP_n$ and current within the parallel strings. A similar behavior is observed during discharge, with string-by-string discharge starting at the potential of the local minimum of the $OCP_n$. Overall, string-by-string charging and discharging results in a horizontal voltage curve. The voltage is determined by the direction of the applied current, showing that the proposed ECM is capable of modeling the voltage hysteresis and representing the system state through the $SOC_n$ within the considered parallel strings.

### Effects of parameter variation

The probability distributed equivalent circuit model (PD-ECM) can be adjusted in a number of ways. These include the definition of the $OCP_n$ function as well as the number, the assigned resistance, resistance range, and shape of the capacity distribution function of the parallel strings.

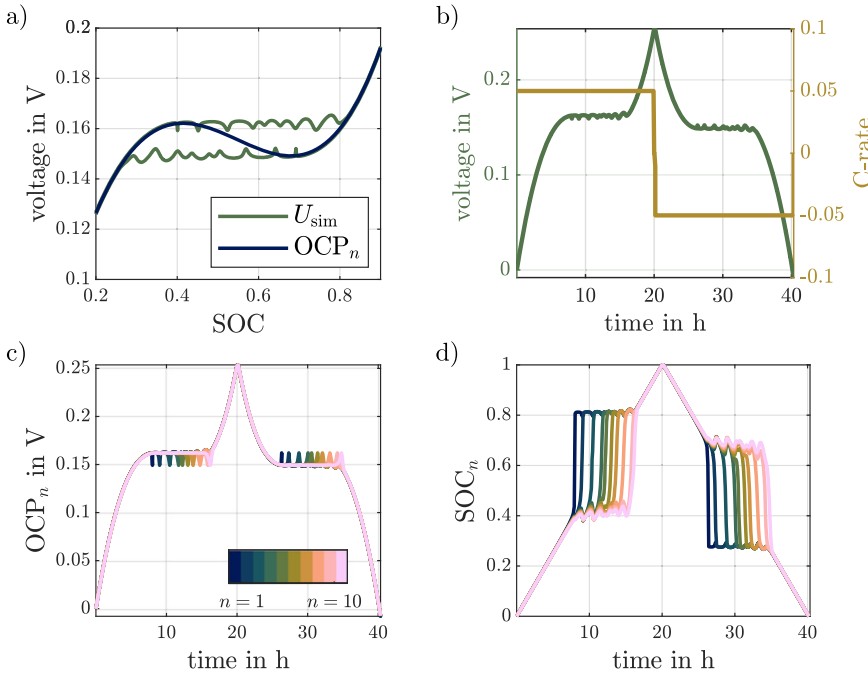

**Fig. 1 | Exemplary simulation of a slow rate full cycle.** Course of a hypothetical single particle open circuit potential ($OCP_n$) and the resulting simulated voltage $U_{sim}$ over the state of charge (SOC) is shown in (**a**), the calculated voltage for the same simulation is together with the input current shown over the time in (**b**), while (**c**) and (**d**) are respectively showing the $OCP_n$ and the single particle state of charge $SOC_n$ for each of the 10 simulated parallel strings over time, from the ones with the lowest resistance ($n = 1$) up to the highest resistance ($n = 10$).

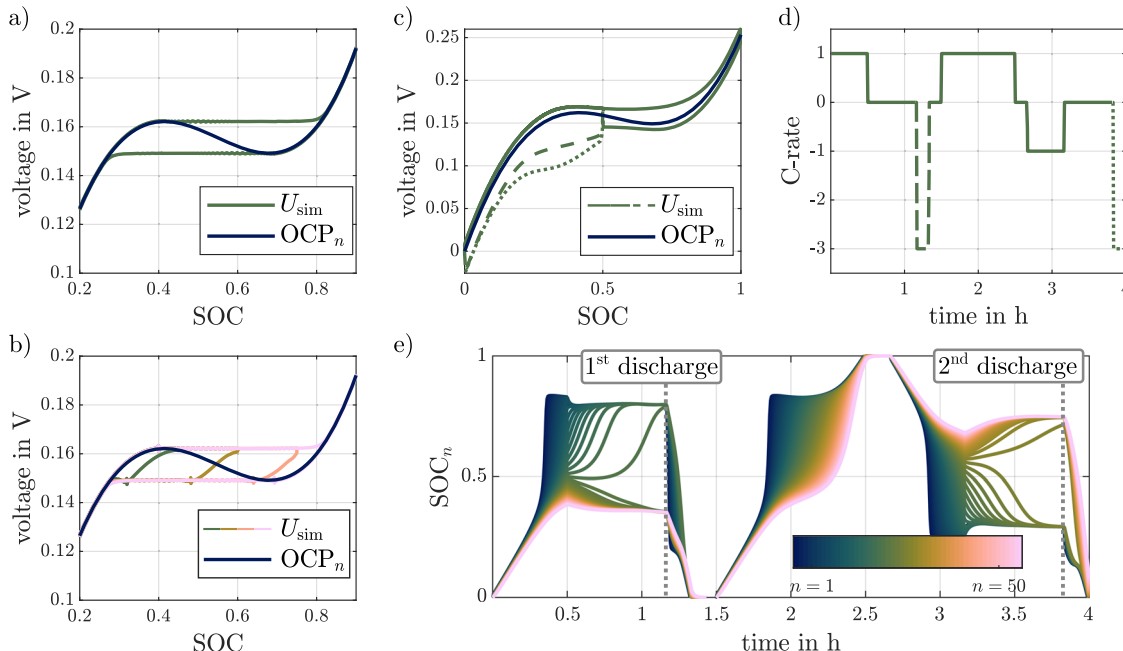

**Fig. 2 | Simulation of slow rate full cycle, partial cycles, and rate capability test using the reference parameter.** Close up of the simulated voltage $U_{sim}$ of a low-rate full cycle together with the single particle open circuit potential ($OCP_n$) in (**a**), simulation of partial cycles together with the $OCP_n$ with visible transition path in (**b**), with each color being a different partial cycle. The voltage during a high-rate discharge beginning at 50% state of charge (SOC) for two different hysteresis states together with $OCP_n$, discharging starting at the upper voltage curve shown with a dashed and at the lower voltage curve with a dotted line in (**c**) with the corresponding current profile of the test in (**d**) and the single particle state of charge $SOC_n$ for the 50 simulated strings in (**e**).

A larger number of parallel strings results in a smaller capacity in each string. One effect of an increasing number of parallel strings is that the simulated voltage in the phase transition region becomes smoother. This can be seen by comparing Fig. 2a with Fig. 1a. The simulation in Fig. 2a uses a PD-ECM with 50 parallel strings, which significantly reduces the height of the voltage ripples compared to the results shown in Fig. 1a where a PD-ECM with only 10 parallel strings is used. However, since the number of states in the state-space representation increases with the number of strings, the computational cost also increases with the square of $N$ (see the Methods section).

Simulation results for the second test, i.e., pseudo OCV test with partial cycles, are shown in Fig. 2b. The test includes partial cycles at low rates between 0% and three different upper SOCs, i.e., 75, 60 and 45%. The simulation results show the expected behavior for the voltage transition when switching from charge to discharge, as a transition between the upper and lower voltage of the full cycle is observed. While phenomenological models implement this behavior explicitly by using inverse exponential functions[20] or quadratic functions[25], this is achieved implicitly here, since in the PD-ECM the voltage transition depends on the state of the parallel strings. Moreover, it can be seen that the slope of the voltage transition

**Fig. 3 | Results of the parameter variation for the rate capability test as well as the discrete distribution of capacity and resistors.** The simulations of the rate capability test for three parameter variations plotted over the cell's state of charge (SOC) are shown in (**a**–**c**). The variation of the resulting resistance $R_{res}$ and the single particle open circuit potential (OCP$_n$) curve is given in (**a**), the close up of the high-rate discharge phase in the simulation for variation of the range of resistances $\Delta R_{norm}$ with OCP$_n$ curve in (**b**), as well as the variation of the shape parameter $k$ of the underlying Weibull distribution in (**c**). In each of the graphs, the dashed line is used for the high-rate discharge starting at the charge curve, while the dotted line corresponds to the high-rate discharge starting at the discharge curve. The plot of resulting shares of capacity for the variation of $k$ depending on the resistance value is given in (**d**).

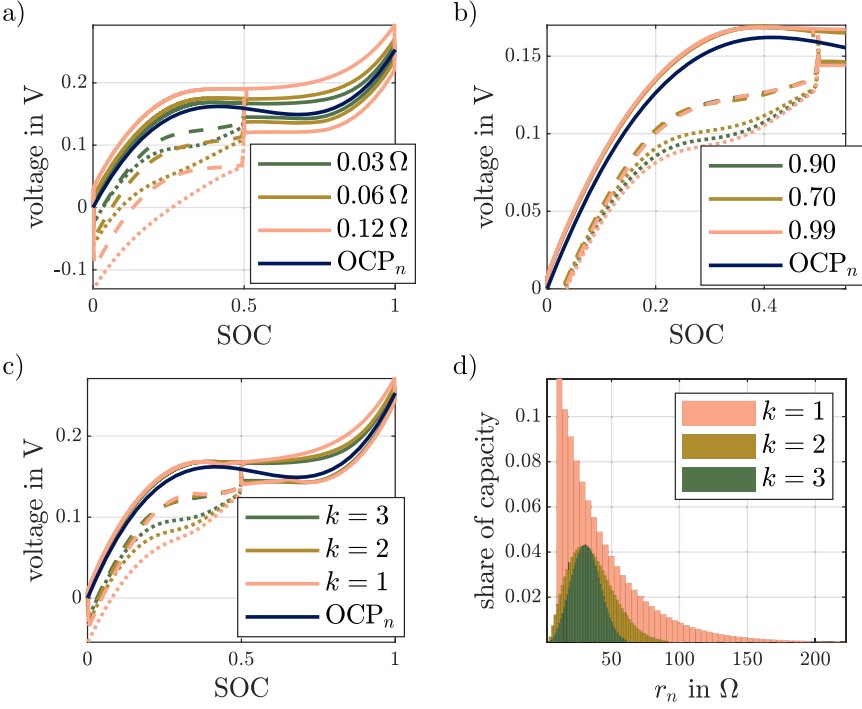

depends on the depth of the partial cycle. For example, the voltage transition for the 75% SOC partial cycle is significantly steeper in the beginning compared to the other partial cycles. Similar changes in the voltage transition are reported for experiments in the literature[37], indicating an advantage of this model approach.

Figure 2c shows the effect of hysteresis on the path dependency test for rate capability. The applied current is shown in Fig. 2d. The rate capability of a cell is determined by the amount of charge or energy that can be extracted from a cell for higher rates. The rate capability is affected by the internal resistance of the cell. To test the effect of path dependency, a constant current discharge of 50% of the cell's capacity is performed for two cases. First, after the cell has been charged to 0.5 SOC. Second, after the cell has been discharged to 0.5 SOC. The first case is shown in Fig. 2c with a dashed line. The second case is shown with a dotted line. The solid line shows the simulated voltage curve for the charge and discharge phases at 1C. Comparing the two cases, it can be seen that the voltage is lower in the second case. This also leads to a decrease in the energy of the discharge step and indicates a dependency of the rate capability on operation history. This path dependency phenomenon is often reported for LFP cells[16–18] and can be represented by the PD-ECM, highlighting another advantage of this model over phenomenological approaches.

The reason for the observed path dependency of the rate capability can be illustrated using Fig. 2e, which shows the SOC$_n$ of the parallel strings. The color of the graphs represents the string number and corresponds to the resistance of the string, with blue corresponding to the lowest resistance and pink to the highest. In the relaxation just before the two high-rate discharge phases, clear differences in the system state can be observed. For the first discharge, the strings with the lowest resistance are at high SOC$_n$. In contrast, for the second discharge, the strings with the highest resistance are at high SOC$_n$. Therefore, in the first case during discharge, low resistance strings are discharged first until all strings reach a common SOC at the local maximum of OCP$_n$. Then all strings are discharged together until they reach 0% SOC. In contrast, in the second case, the strings with the highest resistance are at the higher SOC$_n$ and the strings with low resistance are already at low SOC$_n$. As a result, during discharge, the current flows mainly through the high-resistance strings, resulting in higher polarization, which explains the lower voltage observed. These results show that once the cell is in the

hysteresis region, the strings with the lowest resistance are charged/discharged first. The PD-ECM can represent this dependency of the history of operation by modeling the state in the parallel strings instead of using a lumped SOC of the whole system. As a lumped SOC is common in conventional ECM, they cannot represent this phenomenon.

In several publications, e.g., by Li et al.[38] or Bai et al.[39], it is shown, that the phase separation of LFP can be suppressed under certain circumstances. For nanoparticles charged with high current rates the lithiated iron phosphate changes from particle-to-particle to a concurrent lithiation, forming a quasi-solid-solution or solid-solution. As shown by Katrasnik et al.[40] this leads to very specific behavior during pulse load and the subsequent low load or relaxation phase. The PD-ECM does not incorporate mechanisms aimed to simulate the transition from phase separation to solid solution. However, the behavior shown by Katrasnik et al.[40] that is attributed to the forming of a quasi-solid-solution, can nevertheless be modeled using the PD-ECM without changes in the model structure, as shown in the Supplementary Material (Supplementary Fig. 1). In the PD-ECM the change from particle-to-particle to a concurrent (de)lithiation caused by a high load pulse can be explained by the establishment of a new multi-particle equilibrium, due to the increased voltage drop at the network of resistors. To which extend the results of Katrasnik et al.[40] can be explained by a multi-particle equilibrium and to which by forming of a quasi-solid-solution is not scope of this work and remains a topic for future research.

In the following, the influence of three model parameters are investigated:

(1) the resulting ohmic resistance of the network $R_{res}$.
(2) the range of the normalized resistances $\Delta R_{norm}$.
(3) the function for the probability density of the discrete resistances $p(r_n)$.

For this purpose, these parameters are varied and the results are presented for the path dependency test of the rate capability in Fig. 3. To examine the influence of the shape of the probability distribution function, the Gaussian distribution is replaced by a Weibull distribution, where the parameter $k$ modifies the shape of the distribution while maintaining the distribution width. The parameter values listed in Table 1 are used for the simulations. The analysis is conducted by changing one parameter through all listed values, while the rest is kept at the reference value.

Figure 3a shows simulation results for the variation of the resulting ohmic resistance of the ECM. As can be seen, the polarization increases with higher $R_{res}$ over the whole SOC range. The differences between the simulated voltage curves are slightly higher at higher SOCs, emphasizing that the strings assigned with a high resistance are utilized last. For the high-rate discharges, it can be seen that the polarization is similarly affected by the higher $R_{res}$ for both discharges.

This is in contrast to the influence of the range of normalized resistances $\Delta R_{norm}$ shown in Fig. 3b. Here, the effects are predominantly visible during the high-rate discharge phases. The increased inhomogeneity in the distribution of resistances leads to a significant reduction in rate capability during the second high-rate discharge. For the first high-rate discharge the effects are less significant, since the strings with low resistances are discharged. Consequently, the effect of polarization differences is less significant.

The effect of the shape of the distribution function is shown in Fig. 3c. The corresponding shares of capacity with the corresponding resistance values $r_n$ are shown in Fig. 3d. The visible changes in area covered by the bar plots is caused by changing the distribution. For example, for $k = 1$ a large contribution of smaller resistance values needs to be balanced by few large resistors. This leads to a larger range of $r_n$ in the parallel network to keep $R_{res}$ constant. The sum of shares of capacity over all resistance remains one. Since the resulting ohmic resistance of the PD-ECM is kept constant, comparable overpotentials are observed at the beginning of each charge or discharge

process. As can be seen, the distributions with smaller shape parameters, i.e., a wide distribution of the high-resistance strings, show an increasing polarization when the high resistivity strings are used, i.e., at the end of the charging process. A similar effect can be observed for the high-rate discharges. The simulations with smaller $k$ show a slightly lower polarization during the first discharge due to the higher proportion of low resistance strings. For the second discharge curve a significant influence can be observed. Here, a smaller $k$ leads to significantly increased polarization and thus to a loss of dischargeable energy.

Overall, the parameter variation shows that the model is not only able to simulate the voltage hysteresis during a first-order phase transition, but also allows the parameters to be adjusted to modify the path dependency of the rate capability.

## Half-cell simulation

The next step is setting up the simulation of a cylindrical, commercial A123 battery with LFP cathode and graphite anode using the PD-ECM. The hysteresis of the full-cell depends on the hysteresis of each of the two electrodes. Therefore, each electrode must be simulated separately. To adapt the model parameters to the half-cells, the cylindrical A123 cell is opened, and the electrodes are extracted. These are then used in half-cells to measure the OCV of each electrode. The $OCP_n$ of each electrode cannot be measured directly and therefore must be approximated and parameterized. The whole procedure is described in the Methods section.

The resulting low-rate full cycle for the graphite half-cell with the according error is plotted in Fig. 4a, c, while the LFP half-cell with error is shown in Fig. 4b, d.

As can be seen, the model can be used to simulate the pseudo OCV test including the voltage hysteresis for both electrodes. For both electrodes the error is less than 5 mV for most of the SOC range. However, for LFP there is a significant deviation at very high and very low SOCs. In this range the measured voltage hysteresis is underestimated in the model. This suggests a limitation of the hysteresis model, where the slopes at beginning and end of the plateau are not just implemented but rather a result of the chosen network and the underlying $OCP_n$. However, these observations could also be caused by residual overpotentials in the pseudo OCV measurement that are not correctly accounted for in the model. These could be changes in the

**Table 1 | Parameter values for the reference simulation as well as for the variation simulations with values given for the resulting ohmic resistance of the network $R_{res}$, the range of normalized resistances $\Delta R_{norm}$ and the shape parameter $k$ of the Weibull distribution**

| Parameter | $R_{res}$ | $\Delta R_{norm}$ | $k$ |
|---|---|---|---|
| Reference simulation | 30 mΩ | 0.9 | 3 |
| Parameter variation | 60 mΩ | 0.7 | 2 |
| | 120 mΩ | 0.99 | 1 |

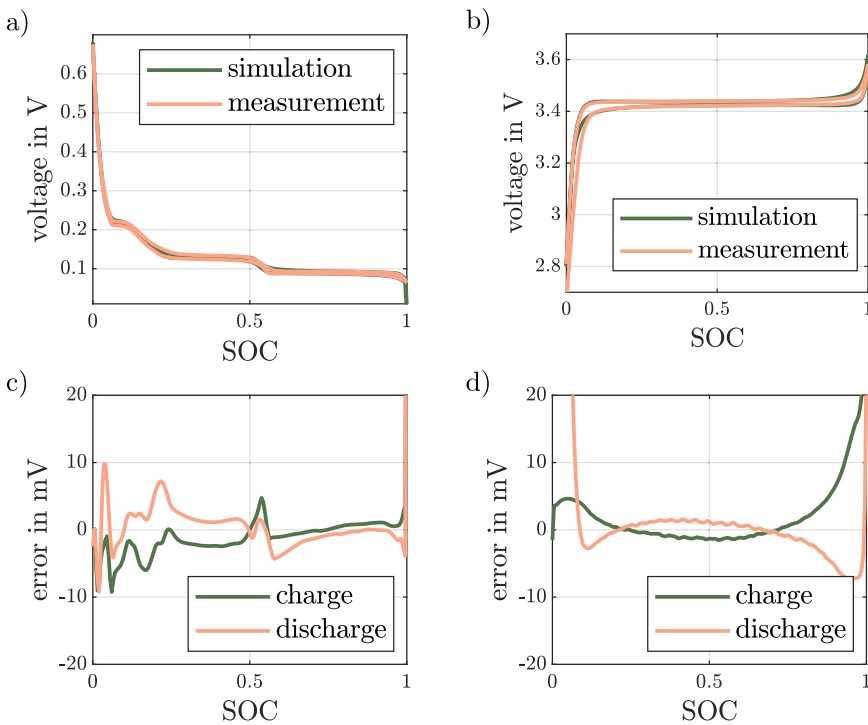

**Fig. 4 | Comparison of measurement and simulation of slow rate full cycles for graphite and lithium iron phosphate half-cells.** Simulation and measurement of the voltage of a graphite half-cell in (**a**) and of a lithium iron phosphate half-cell in (**b**) plotted over the cell's state of charge (SOC) with the deviation of the simulation from the measurement for graphite in (**c**) and for lithium iron phosphate in (**d**).

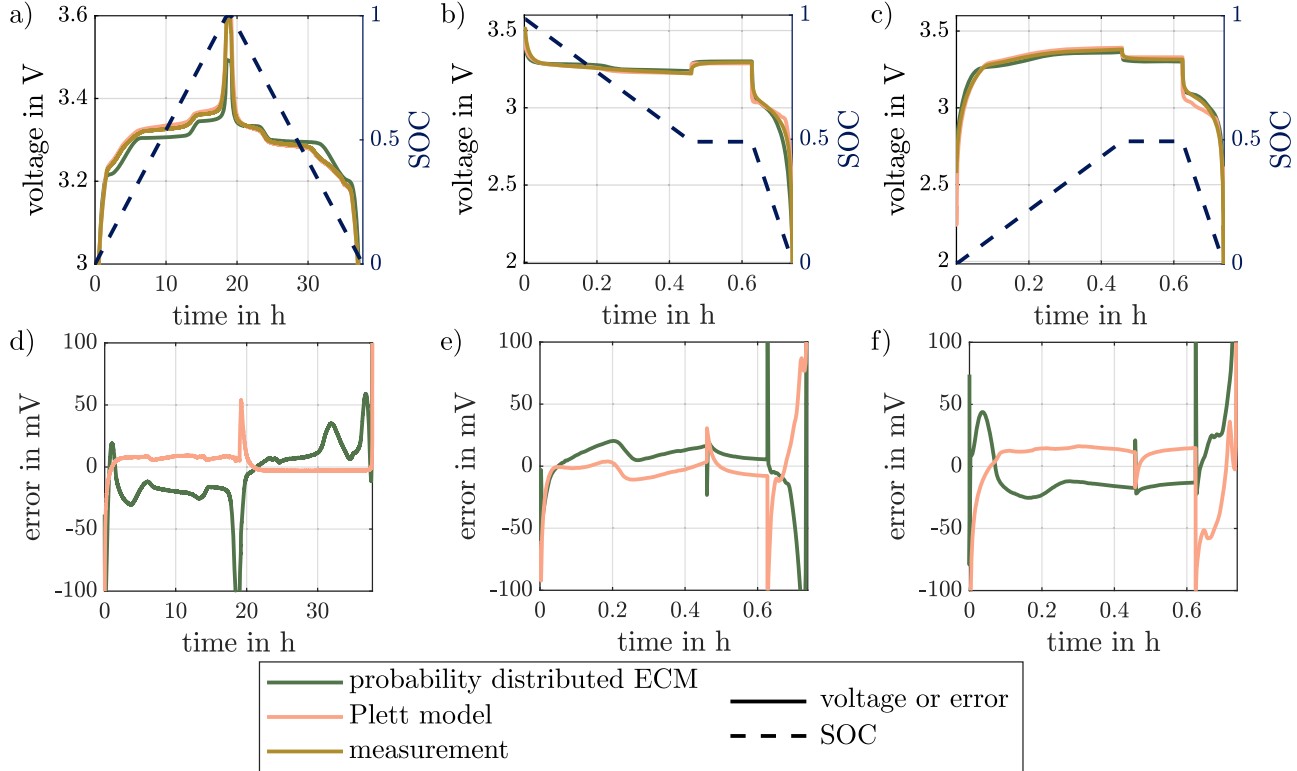

**Fig. 5 | Simulation results for a slow rate full cycle and rate capability test for a commercial lithium-ion battery.** Voltage over time for the probability distributed equivalent circuit model (ECM) and the Plett model, as well as the measurement results for a low-rate full-cycle in (**a**) with the error in (**b**), for a rate capability test starting on the charge curve in (**c**) with the error in (**d**), and the rate capability test starting on the discharge curve in (**e**) with the error in (**f**). For (**a**–**c**) the cell's state of charge (SOC) is plotted on the second *y*-axis.

charge transfer resistance or diffusion overpotentials not considered in the half-cell simulation using the PD-ECM.

## Full-cell simulation

For the full-cell simulation the half-cells are simulated separately and the results are subtracted. In addition, a serial connection of two RC elements is added to the model to account for further polarization. The ohmic resistor connected in serial that is commonly used in ECM for batteries is replaced by the distributed resistors in the PD-ECM. These further model modifications and model parameterization are described in the Methods section.

To evaluate the performance of the model, the results are compared with an established hysteresis model from the literature. The commercial LFP cell from A123 is used for the measurement. Figure 5 shows the results for three current profiles. Figure 5a shows the results for the pseudo OCV test. Figure 5b, c depicts the results for the path dependency test for rate capability. The corresponding errors of the simulated voltages are shown in Fig. 5d–f. The measured current is used as the input for the simulations. The simulation results of the PD-ECM are compared with the single-state hysteresis model proposed by Plett[20]. The implementation and parameterization of the single-state hysteresis model is described in the Methods section. The model uses the same SOC-dependent parameters for the preconnected RC-model as the PD-ECM and differs in the modeling of the voltage hysteresis and an additionally implemented serial resistor. The results of additional simulations of partial cycles with low-rate are given in the Supplementary Methods in Supplementary Fig. 2.

The pseudo OCV test and the corresponding errors of both models are shown in Fig. 5a, d respectively. As can be seen, the error of the single-state hysteresis model is with a relative root mean square error (RMSE) of 0.432% only half of the RMSE for the PD-ECM with 0.9323%. This is not surprising as the measured data from the pseudo OCV test is fed directly into the model as the upper and lower OCV curves. The visible errors are only caused by

polarization errors, which are very small for the low C-rates used. Nevertheless, the PD-ECM results are of sufficient precision, when comparing with the reported RMSE for the simulation of LFP using a Preisach model by Baronit et al.[23] ranging from 0.6% to 2.5%. Still, the error for the full-cell simulation is significantly higher than the sum of the errors of the half-cell measurement. This indicates a problem arising from the scaling from half-cell to full-cell measurement. When taking a closer look at the plateaus following the stage transition of graphite two different slopes for the PD-ECM simulation and the full-cell measurement are visible. This may be an indicator for more inhomogeneous lithiation in the full-cell compared to the half-cell. Due to the parameterization of the PD-ECM using half-cells these inhomogeneities are not taken into account. This problem is avoided in the single-state hysteresis model as it can be directly parameterized with full cell measurements.

The results of the rate capability tests are shown in Fig. 5b, c with the corresponding errors in 5e, f. For the first test the 0.5 SOC was set from a fully discharged state (Fig. 5b), while for the second test the 0.5 SOC was set from a fully charged state (Fig. 5c). In both cases the analyzed models have comparable errors during the 1C charge or discharge phases. However, during the high-rate discharge the PD-ECM shows an advantage. The single-state hysteresis underestimates the voltage in the first test and overestimates the voltage in the second test. The PD-ECM can account for the path dependency of the rate capability and follows the slope of the measured voltage better. This also results in smaller errors at the beginning of the discharge. However, at the end of the discharge, both models have significant errors, which may be due to insufficient modeling of the cell polarization. The RMSE of the PD-ECM simulation for the rate capability test on the charge curve is with 2.856% and with 1.97% for the rate capability test on the discharge curve significantly higher than the slow rate simulation. Comparing these to dynamic tests and ECM simulation conducted by Guenther et al.[25], who reach a median deviation of 0.06%, shows worse final

**Fig. 6 | Working principle and structure of the probability distributed equivalent circuit model.** Scheme of the basic principle of the hysteresis model with string resistors $R$ and the corresponding capacity elements $\Delta Q$ in (**a**) with an exemplary probability $p(r)$ distribution of the resistances $r$ in (**b**) and the resulting equivalent circuit model with charge controlled voltage sources following the single particle open circuit potential (OCP$_n$) in (**c**). The voltage drops $U_{cl}$ and $U_{r,1}$ and the current $I_0$ are exemplary depicted as well. The qualitative depiction of the OCP$_n$ over the single particle state of charge (SOC$_n$) in blue and the expected open circuit potential (OCP) curve for the whole cell over its state of charge (SOC) for a full cycle of the network in gold are shown in (**d**).

results for the PD-ECM. Nevertheless, the simulation error for the dynamic test cases is still comparable to the simulation of Baronti et al.[23]. Since Guenther et al. did not simulate the edge regions of the SOC, where the largest model errors are occurring, this may explain the different results.

## Conclusion

The presented results show the capability of the probability distributed equivalent circuit model comprising a non-monotonous potential curve OCP$_n$, controlled by the state of charge in each string SOC$_n$, to simulate the hysteresis behavior of materials undergoing a first-order phase transition. For the first time, the physical principles of hysteresis were successfully transferred into an ECM. The additional model parameters allow for a detailed tuning of the voltage behavior under load, including hysteresis effects often neglected in conventional modeling approaches for BMS application, such as rate capability and power prediction. The model design shows some disadvantages as well: a low number of simulated parallel strings lead to high ripples in the simulated voltage curve. Moreover, the system does not prove stable for all conditions, especially for high values of $\Delta R_{norm}$. Due to the need to use half-cell data to obtain a matching full-cell voltage simulation, the effort to set up the model is comparatively high. In addition, approximating the OCP$_n$ curve along with the set of optimal model parameters is a challenging optimization task due to its non-linearity. This problem is exacerbated when the two half-cell simulations are to be combined for the full cell. Establishing a workflow for the OCP$_n$ curve approximation and parameter validation for full-cells is therefore a topic for further research.

## Methods

In this section, the mathematical description of the state-space representation of the PD-ECM is derived. This includes the description of the governing equations and the solution algorithm. In addition, the implementation of an established hysteresis model is described, the test procedure, and the parameterization workflow is presented.

## Model idea

The ECM hysteresis model is based on the model introduced by Dreyer et al.[5]. They show that in a multi-particle system, particle-by-particle charge occurs, if a non-monotonic potential curve is applied. In the work of Kondo et al.[33] it was further shown that by using a distributed particle size the sequence of the phase transition is determined and the path dependency of the rate capability can be described. They assumed that the differences in charge transfer resistance caused by varying particle size is the dominant influence on the sequence of particle-by-particle lithiation, while electrical and ionic conductivity are negligible. Thomas-Alyea[41], though, proposed a resistive-reactant model accounting for holes in the conductive matrix around LFP particles as reason for inhomogeneous particle lithiation. EDXRD measurements by Paxton et al.[42] showed good agreement with the model prediction. To account for both effects the PD-ECM uses distributed

resistances with a constant particle size. While an additional variation of the particle size is closer to the actual electrode structure, particle size variation is only represented by its influence on the resistance in the PD-ECM. In this work the concepts of non-monotonic potential curves and a variation of the particle resistance are combined in an equivalent circuit model, which is the most common modeling approach used in BMS.

In Fig. 6a, the underlying idea of the PD-ECM is illustrated. It shows the network of $N_{ini}$ partitions of capacity:

$$\Delta Q = C/N_{ini}, \tag{1}$$

with the overall cell capacity $C$ in connection with a specific resistor for each string. The parallel resistances add up to the overall resistance $R_{res}$ by:

$$R_{res} = \left( \sum_{i=1}^{N_{ini}} \frac{1}{R_i} \right)^{-1}. \tag{2}$$

The resistors are assigned to $N$ bins, which in turn represent uniformly distributed resistance values in the fixed range $[R_{min}, R_{max}]$. The probability function $p$ defines the frequency of each discrete resistance value in the network. Exemplary, the probability density of a normal distribution is shown in Fig. 6b. Here, the probability of being assigned to bin $R_2$ is three times higher compared to $R_1$. Looking at Fig. 6a this would translate into $R_I$ belonging to bin $r_1$, while $R_{II} = R_{III} = R_{IV}$ are all assigned to bin $r_2$.

The model idea is translated into an ECM, as depicted in Fig. 6c. The ECM consists of $N_{ini}$ resistors connected in parallel and a charge controlled voltage source in each string, representing the partition of capacity. For the hysteresis model to work as described above, a non-monotonous potential curve is assigned to the voltage sources. The potential OCP$_n$ of each voltage source is depending on the amount of charge stored in each parallel string relatively taken to the assigned capacity $\Delta Q$. An exemplary OCP$_n$ curve over the SOC$_n$ of each string is shown in Fig. 6d, as well as the expected voltage curve for the whole network, depending on the direction of the current.

## Model implementation

At first, a uniformly distributed vector of resistance values is defined. The length of the vector is determined by the number of bins $N$. The maximum and minimum values of the vector are in relation to the center of the interval defined to be:

$$R_{min} = R_{center} - \Delta R \tag{3}$$

$$R_{max} = R_{center} + \Delta R. \tag{4}$$

$\Delta R$ is constrained to $\Delta R \in (0, R_{center})$ to prevent negative values.

Now, the relative frequencies $P_n$ of the resistors associated with each bin are calculated. The frequency is given by the area under the curve of the

probability density function. The function of the probability density $p(r)$ as a function of continuous resistance values $r$ must be discretized to obtain a value for each bin. For the Gaussian distribution, with the probability density function:

$$p_G(r) = \frac{1}{\sigma\sqrt{2\pi}} e^{-\frac{1}{2}\frac{r-\mu^2}{\sigma}}, \quad (5)$$

this is done by limiting the function to $3\sigma$ and splitting the range into $N$ uniformly sized intervals:

$$\delta r = \frac{2 \cdot \Delta R}{N}. \quad (6)$$

With the resistance vector and the definition of $\Delta R$ the expectation value and standard deviation of the Gaussian distribution are $\mu = R_{center}$ and $\sigma = 1/3 \cdot \Delta R$, respectively. The relative frequency of resistors in each bin is then given by:

$$P_{G,n} = \int_{R_{min}+(n-1)\cdot\delta r}^{R_{min}+n\cdot\delta r} p_G(r)\, dr. \quad (7)$$

For the analysis of the influence of changes on the distribution function, a Weibull distribution is implemented. The Weibull distribution is defined by the shape parameter $k$ and the inverse scaling parameter $\lambda$. The probability density function is given by:

$$p_W(r) = \lambda \cdot k \cdot (\lambda \cdot r)^{k-1} e^{(\lambda \cdot r)^k}. \quad (8)$$

Similar to the Gaussian distribution the relative frequency of resistors in each bin is calculated by:

$$P_{W,n} = \int_{R_{min}+(n-1)\cdot\delta R}^{R_{min}+n\cdot\delta R} p_W(r - R_{min})\, dr. \quad (9)$$

The last design parameter of the PD-ECM is the $OCP_n$. The OCP of the $n$th string can be any non-monotonous analytical function of the $SOC_n$. In this publication the $OCP_n$ is defined as:

$$OCP_n(SOC_n) = \left((1 - \gamma(SOC_n)) \cdot y_1 + \gamma(SOC_n) \cdot y_2\right) \cdot SOC_n \quad (10)$$

$$\gamma(SOC_n) = -\left(\cos\left(\pi \cdot \frac{SOC_n - x_1}{x_2 - x_1}\right) - 1\right). \quad (11)$$

For the model introduction and parameter variation the variables of Eqs. (10) and (11) were chosen to be $x_1 = -0.8$, $x_2 = 0.8$, $y_1 = 1.6$, and $y_2 = 0.9$.

In the Results and discussion section $\Delta R$ is given normalized to the center value of the vector of resistances to improve comparability between results. Therefore, $\Delta R_{norm} = \Delta R/R_{center}$ is used in the range from (0, 1).

**Model reduction**
The model described above is reduced to improve the performance of the simulation. This is done by pooling resistors, i.e., strings with the same resistor get combined. The relative frequency $P_n$ of each resistance value is multiplied with the total number of resistors $G$. The resulting resistance $R_n$ of a number of $N_P = P_n \cdot G$ equal resistors with resistance $r_n$ connected in parallel is:

$$R_n = r_n \cdot \frac{1}{N_P}. \quad (12)$$

Consequently, the capacity assigned to the bin is:

$$C_n = \Delta Q \cdot N_P. \quad (13)$$

**Table 2 | Ratio between calculation time $t_{calc}$ and simulation time $t_{sim}$ for a 0.05C full-cycle depending on the number of parallel strings $N$**

| $N$ | 10 | 30 | 50 | 75 | 100 | 125 | 150 | 200 |
|---|---|---|---|---|---|---|---|---|
| $t_{calc}/t_{sim}$ in ‰ | 0.097 | 0.23 | 0.48 | 1.1 | 1.7 | 2.3 | 3.3 | 7.8 |

With Eq. (12), the resulting resistance of the network is:

$$R_{res} = \left(\sum_{n=1}^{N} \frac{1}{R_n}\right)^{-1}. \quad (14)$$

Model reduction is crucial to the applicability of the PD-ECM in a BMS environment. The electrode's physical properties with thousands of particles per mm$^2$ result in an almost continuous variation of resistivity values in the electrical pathways. Binning these values simplifies the network but still involves high computational cost. Reducing the model to fewer equivalent resistances and capacities allows fast calculation while preserving the basic assumptions of the model. For example, by reducing the number of elements from 1000 to 50, the calculation time is reduced by approximately 400, with a quadratic decrease as illustrated in Table 2.

**Solving algorithm**
For the full-cell model two RC elements are added in series to the hysteresis ECM to model the dynamic behavior of the battery. The resulting differential-algebraic system of equations (DAE) is given in its state-space representation. The state vector $\mathbf{x}$ is defined as:

$$\mathbf{x}_k = \begin{bmatrix} U_{RC1} & U_{RC2} & I_1 & \dots & I_{n-1} & Q_1 & \dots Q_n \end{bmatrix}^\top \quad (15)$$

with the voltage drop over each RC element $U_{RC1}$ and $U_{RC2}$ respectively, the line currents $I$ and the stored charge in each string $Q$. The discrete-time form of the state-space representation with algebraic equations and isolated non-linearities is:

$$\mathbf{M}\mathbf{x}_{k+1} = \mathbf{A}\mathbf{x}_k + \mathbf{b}u_k + \mathbf{v}_1(\mathbf{x}). \quad (16)$$

with the system response $y_k$ given by:

$$y_k = \mathbf{c}\mathbf{x}_k + du_k + \nu_2(\mathbf{x}). \quad (17)$$

The definition for each matrix and vector is given in the Supplementary Methods in Eqs. (S.1)–(S.5). With reference to Hahn et al.[43] the system of equations is solved with a linear-implicit Euler method:

$$(\mathbf{M} - h \cdot \mathbf{f}'(\mathbf{x}_k, t_k))(\mathbf{x}_{k+1} - \mathbf{x}_k) = h \cdot \mathbf{f}(\mathbf{x}_k, t_k) \quad (18)$$

with the time step $h$, and the system function $\mathbf{f}$. The utilization of the LU-factorization enables the necessary inversion of the matrix:

$$\mathbf{J} = \mathbf{M} - h \cdot \mathbf{f}'(\mathbf{x}_k, t_k) \quad (19)$$

without iterative calculations, assuring stable computation times for solving the DAE. To show the real-time capability of the algorithm, the simulation of a slow rate full cycle with more than 40 h simulation time is conducted on a PC with Intel i7-10510U processor and 16 GB RAM. By changing $N$, the influence of the number of parallel strings on the calculation time $t_{calc}$ is demonstrated. The results of the study are given in Table 2, showing an increasing calculation time with the square of $N$. Even for large numbers of $N$, the calculation time is two magnitudes smaller than the simulation time.

## Single-state hysteresis model

Plett et al.[20] derive their model for the dynamic hysteresis voltage from a function $f(\text{SOC}, t)$ of SOC and time as well as the maximum polarization $M(\text{SOC}, \dot{\text{SOC}})$ that has its origin in hysteresis. With the simplification of $M(\text{SOC}, \dot{\text{SOC}}) = M \cdot \text{sign}(I)$ and further mathematical operations the difference equation:

$$x_{\text{hys},k+1} = \exp\left\{-\left|\frac{\eta \cdot I_k \cdot \gamma \cdot h}{Q}\right|\right\} \cdot x_{\text{hys},k} \\ - \left(1 - \exp\left\{-\left|\frac{\eta \cdot I_k \cdot \gamma \cdot h}{Q}\right|\right\}\right) \cdot \text{sign}(I_k) \tag{20}$$

for the unitless hysteresis state $x_{\text{hys}}$ is derived. The rate of alteration in the hysteresis voltage is linked to how far the current hysteresis value deviates from the main hysteresis loop. This results in an exponential decline of voltage toward the primary loop. The positive constant $\gamma$ controls the pace of this decline. Additionally to the dynamic hysteresis that changes with SOC, instantaneous hysteresis can be modeled by $M_0 \cdot s_k$, with:

$$s_k = \begin{cases} \text{sign}(I_k), & \text{for } |I_k| \geq 0; \\ s_{k-1}, & \text{otherwise}. \end{cases} \tag{21}$$

Equation (20) can easily be implemented in a state-space representation by including $x_{\text{hys}}$ as an additional state. Together with $M$, $x_{\text{hys}}$ and an SOC-dependent mean OCV curve the resulting hysteresis-affected voltage is described by:

$$U_{\text{hys},k} = M_0 \cdot s_k + M \cdot x_{\text{hys},k} + \text{OCV}_k \tag{22}$$

and is added to the output equation of the state-space representation.[44]

## Full- and half-cell measurements and tests

For the validation of the hysteresis model a cylindrical A123 18650M1A cell in the format 18650 is chosen. The cells consists of a graphite anode and a LFP cathode and therefore show a significant hysteresis dominated by first-order phase transitions. All tests are carried out using a BaSyTec CTS battery tester in a Binder MK115 climate chamber at 25 °C. After preconditioning the cell with ten 1C full-cycles, it undergoes a two-part testing procedure: first, the cell's characterization, and second, the conduction of test cases for the validation of the hysteresis model.

The cell's characterization involves conducting pulse tests to determine the parameters of the R-2RC model. Each pulse test consists of the following steps: The cell is charged to a specific SOC and allowed to rest for 2 h. Afterwards, a 1C, 10 s charge pulse is applied to the cell and its relaxation behavior is observed for 1 h. This procedure is repeated for every 1% SOC step between 0% to 5% and 95% to 100% SOC, as well as for every 5% SOC increment between 5% and 95%.

The first test case for the model validation is a full cycle with a low current rate of 0.05C. To analyze the transition behavior from the charging hysteresis curve to the discharging curve, expanding hysteresis loops starting at 50% SOC are measured in the second test. Finally, the influence on the path dependency of the rate capability is demonstrated by simulating a fast discharge with 3C at 50% SOC once starting on the charge voltage curve and once starting from the discharge voltage curve.

For the determination of the OCPs of the electrodes, the cell is opened after the tests. Samples of the electrodes are punched out and assembled in EL-Cell PAT cells with lithium metal counter electrode, glass fiber separator, lithium reference electrode, and LP57 electrolyte. The cells are rested for 24 h in a temperature chamber at 25 °C. Afterwards a BaSyTec CTS-LAB is used to cycle the cells three times with 0.05C to assure a stable cell setup. The OCP is determined by performing a galvanostatic intermittent titration technique (GITT) test as well as a slow rate full cycle with 0.05C with both electrodes. In the GITT for each 1% SOC step, the potential measured at the end of the 10 min rest phase is read out as OCP.

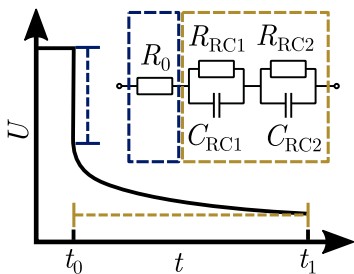

**Fig. 7 | Depiction of the preconnected R-2RC model with an illustration of the voltage response during rest and the ranges used for parameterization.** The qualitative voltage response $U$ over time $t$ during the cell's relaxation is shown with the part of the signal used for parameterization of the ohmic resistance $R_0$ marked blue. The part of the signal used for the parameterization of the two resistor-capacitor elements (resistors $R_{\text{RC}}$ and capacitors $C_{\text{RC}}$) in the preconnected equivalent circuit model is marked with a yellow horizontal line.

## Parameterization

First step of the parameterization of both models is the determination of the SOC-dependent values of the R-2RC model, depicted in Fig. 7. Since the same preconnected elements are used in both models, the parameterization procedure is identical. For the determination of the values the rest phase after each charge pulse in the pulse characterization is used. At first the difference between the last data point of the pulse and the first of the rest phase is divided by the current amplitude $I_0$ of the pulse to calculated the resistance $R_0$ of the processes with very low time constants, as shown in Fig. 7. Afterwards, the optimization problem:

$$\min f(R_{\text{RC1}}, C_{\text{RC1}}, R_{\text{RC2}}, C_{\text{RC,2}}) = U_{\text{meas}} \\ -(I(t) \cdot R_0 + I_0 R_{\text{RC1}} \cdot e^{-\frac{t}{R_{\text{RC1}} C_{\text{RC1}}}} + I_0 R_{\text{RC2}} \cdot e^{-\frac{t}{R_{\text{RC2}} C_{\text{RC2}}}}) \tag{23}$$

is solved using a least-squares optimization algorithm (Matlab's *lsqnonlin* function). $U_{\text{meas}}$ is the measured voltage from $t_0$ to $t_1$ as indicated in Fig. 7. The resulting parameter used in both models are given in the Supplementary Methods in Supplementary Table 1.

Since the $\text{OCP}_n$ curve of the materials cannot be directly measured, it must be approximated. For both electrodes the incremental OCP is taken and the first derivative used to determine the SOCs of the phase transitions. Afterwards Eq. (10) is used to model the non-monotonous part of the $\text{OCP}_n$ curve. $x_1$ and $y_1$ are taken at the start of the phase transition from the upper voltage curve while $x_2$ and $y_2$ are taken from the lower voltage curve. The part of the curve between the transition areas is interpolated using a piecewise cubic Hermite interpolating polynomial (Matlab's *pchip* function). Afterwards a least-squares optimization approach for nonlinear problems (Matlab's *lsqnonlin* function) is used to identify the optimized parameter of Eq. (10). In our simulations, $N = 50$ turned out to be a satisfactory compromise between calculation speed and ripples in the voltage. $R_{\text{res}}$ was oriented at the measured $R_0$ values determined in the pulse characterization. $\Delta R_{\text{norm}}$ and the shape parameter $k$ of the Weibull distribution are varied in the range between 0.7 and 0.999 for $\Delta R_{\text{norm}}$ and between 1 and 3 for $k$. For the full-cell simulation the values are chosen to represent the behavior during the rate capability test with small error. Since both electrodes are simulated separately, for the simulation of the commercial full-cell the electrode balancing is needed. The balancing is obtained by analyzing the position of stage transition peaks in the differential voltage of the full cell. These peaks are then aligned with the features in the electrodes.

For the Plett hysteresis model, $M$ is computed directly as the difference between the mean OCV and its respective charging and discharging curves. The deviation is averaged over both instances and the values are stored at each SOC sampling point. This is not the standard procedure proposed by Plett[44], but yields the results with the smallest deviations for our application. Finally, the value for $\gamma$ is optimized within a range of 0 and 300 until the

smallest deviation between a modeled and measured OCV loop is reached. In this publication 25 is identified as an optimal result for this parameter.

## Data availability

The data that supports the findings of this study are available on Zenodo (https://doi.org/10.5281/zenodo.10852930).

## Code availability

The code for the model that supports the findings of this study is available on Zenodo (https://doi.org/10.5281/zenodo.10852695).

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

## Acknowledgements

The authors would like to thank Srivatsan Ramasubramanian from the Bavarian Center of Battery Technology (BayBatt) for the scientific discussion and critical review of the manuscript. Furthermore, the authors thank the graduate school of the BayBatt for ongoing support. Support by the BayBatt Cell Technology Center is gratefully acknowledged, funded by the Deutsche Forschungsgemeinschaft (DFG, German Research Foundation)—INST 91/452-1 LAGG. The color palette used in this report was developed by Crameri et al.[45] and is publicly available on Zenodo (https://doi.org/10.5281/zenodo.8035877). This work is funded by the German Federal Ministry of Education and Research (BMBF) under funding reference 03XP0321A.

## Author contributions

Leonard Jahn: methodology, software, investigation, data curation, validation, writing—original draft, visualization. Patrick Mößle: methodology, software, validation, writing—original draft. Fridolin Röder: conceptualization, methodology, writing—review & editing, supervision. Michael A. Danzer: methodology, writing—review & editing, supervision, funding acquisition.

## Competing interests

The authors declare no competing interests.
