## [Peer Review File · Communications Engineering]

Reviewers' comments:

Reviewer #1 (Remarks to the Author):

This paper presents a probability distributed equivalent circuit modelling approach to capture the voltage hysteresis effect observed in battery electrode materials. The work is interesting, relevant to readers of Communications Engineering and well executed. However there are a number of comments of note for consideration:

- The work presented highlights graphite, LFP and silicon are notable materials which exhibit this voltage hysteresis effect. There is some discussion in the literature review which then briefly mentioned the mechanisms of this. Here a bit more detail would be useful. However, the main point here is that the model construction seems to be centred around creating a framework for capturing the behaviour of phase separating materials such as LFP. As the authors mentioned, silicon is a bit different in that, whilst it does have a significant hysteresis, many comment on this coming about from different silicon phases (amorphous, crystalline etc) which arise during (de)lithiation. This is a bit different to the LFP behaviour and I suppose you want to be clear if you think this framework is applicable for silicon or not.

- When varying the C-rate there are some interesting behaviours. Perhaps I missed this when reading, however I think the core framework of the model at the higher rates in the same (i.e. capturing these different co-existing phases). In the various works by Bazant et al. they have done extensive work on how this phase separation can be suppressed in LFP at higher rates. Here is an example paper (<https://pubs.acs.org/doi/10.1021/nl202764f>). It would be useful to contrast your work with that of this general phase separation suppression behaviour observed at higher rates leading to a quasi solid solution behaviour.

- The path dependency work is also interesting and somewhat following up on the previous point, it would be useful to see if your modelling framework captures the recently reported effects around OCV recovery after high current pulses shown here (<https://onlinelibrary.wiley.com/doi/full/10.1002/adma.202210937>). Here, this interesting behaviour arises due to this claimed quasi-solid-solution behaviour, which I don't believe your framework includes. This is fine of course, however it would be a useful reflection in the limitations of the modelling approach.

- On page 10, lines 279-283, the authors comment on the loss in accuracy at low and high SOCs, with this being attributed to residual overpotentials. It would be useful to unpack this a bit more in terms of the origins of this behaviour and this statement, as one might expect the overpotential to be quite low at low currents used to simulate the pseudo-OCV.

Minor points

- Define SOC on first use in main text

- You mention current and voltage clamps. I understand what you mean here however perhaps this is overtly specific and you would be better just to refer to the voltage and current for more physically meaningful text

- When discussing the voltage hysteresis on page 2 line 17-18 it would be useful to give a relative magnitude to give readers a sense of this, especially when you later present modelling accuracy in mV

- On page 2 line 30, the authors mention that the OCV is defined by the charge throughput, whilst this is

somewhat true, it might be more accurate to say it is based on an electrodes state of lithiation as charge throughput will be dependent on the capacity of the cell

- On page 4 line 128 and 131 the authors mention a “horizontal course” and “time course”. This could be rephrased to make it more physically meaningful and clear

- In various places throughout the manuscript, the authors comment on “high” and “low” behaviour. It would be useful to quantify this a bit more for greater scientific accuracy

Reviewer #2 (Remarks to the Author):

This paper establishes a probability distributed equivalent circuit model (PD-ECM) motivated by the physical insight into hysteresis, the behavior of which is elaborated and validated with commercial cylindrical cells. The manuscript presents some significant results worthy of publication but should be revised to be more precise and accurate in the wording of the text. I want to put forward the following suggestions to improve the quality of the paper.

(1) The novelty of the work should be explained in detail in the Abstract, Highlights, and Conclusion. The main content should concern your research idea and its experimental effect verification.

(2) The expression of the abstract should be improved. A more detailed presentation of innovation should be conducted as well as the experimental verification.

(3) More recent literature should be cited and analyzed, such as Improved anti-noise adaptive long short-term memory neural network modeling for the robust remaining useful life prediction of lithium-ion batteries, Improved singular filtering-Gaussian process regression-long short-term memory model for whole-life-cycle remaining capacity estimation of lithium-ion batteries adaptive to fast aging and multi-current variations, and so on.

(4) The expression logic can be improved for your proposed method so that the innovation can be clearly understood. More expression of the mathematical analysis should be conducted to show your ideas clearly. Also, please try to highlight your proposed method and focus on it.

(5) Grammar should be checked and improved for the entire content. Please try to make every sentence to be correct and easy to be understood.

(6) More experimental result comparisons with references should be conducted for advantage discussion.

(7) Many contents are expressed for the traditional methods for this version. Please pay more attention to your ideas and innovation. Also, please describe your invention briefly for the Conclusion.

(8) The whole structure should be improved. For this version, there is so much content expressing your experiments and basic principles, but not enough expression for your particular idea and methods.

Reviewer #3 (Remarks to the Author):

itle: A physically motivated voltage hysteresis model using a probability distributed equivalent circuit

In this article, the authors describe a modeling approach to model the voltage hysteresis of first order

phase transition materials like LFP. They use a physically motivated ECM to describe not only the voltage hysteresis but also the history-dependency of the rate-capability of LFP electrodes.

As the hysteresis effect in LFP cells is relevant for SOC estimation and understanding LFP-specific phenomena, the paper is on a topic of relevance and general interest to the readers of the journal.

The article is well written with a good flow and most of the important aspects are given.

However, it needs a major revision before being eligible for publication in communications engineering. The following issues should be addressed:

1. As you showed in chapter 3, Kondo et al. already developed a model with consideration of non-monotonic eq. potential curves. As they also do not consider diffusion in solid and liquid phase, this model can easily be transferred into an ECM. So please explain the novelty of your approach more in detail compared to the model of Kondo et al..
2. Following on from question 1., in your model description (fig. 6) you assign the same capacity to all particles and you just change the ohmic resistance which is connected in series to the voltage source. Please give more detail why you chose this way and not a distribution of capacity and resistance according to a fitted (gaussian distributed) particle-size distribution (similar approach to Kondo), which would be more physically motivated. Would that add computation-time?
3. Please explain more in detail how you parameterized the distribution of resistances. What measurements were used to fit the standard deviation of the distribution? How did you separate these resistances for the current distribution from the series resistance of the cell to not account it twice in the model?
4. Is the Plett-model simulated for each electrode individually? If not, would the results change (fig. 5)?
5. Considering the deviation between the PD-ECM parameterized on half-cell level and the measurements on full-cell level, could those differences arise due to inhomogeneities in the full-cell? Have you done full-cell measurements in experimental cells to rule out errors due to inhomogeneities which you are not able to attribute with your 2RC-ECM? I am suggesting this, as you see a sloping behavior of the measurement in the plateau regions (more inhomogeneities) and smaller errors in the phase transition regions (smaller inhomogeneities in the cell).

We thank the reviewers for their valuable feedback that helped us to further strengthen our manuscript. Below you can find a point by point response to the remarks of the reviewers. Our answers are indicated in red.

Revision „A physically motivated voltage hysteresis model using a probability distributed equivalent circuit“

Reviewer #1 (Remarks to the Author):

This paper presents a probability distributed equivalent circuit modelling approach to capture the voltage hysteresis effect observed in battery electrode materials. The work is interesting, relevant to readers of Communications Engineering and well executed. However there are a number of comments of note for consideration:

- The work presented highlights graphite, LFP and silicon are notable materials which exhibit this voltage hysteresis effect. There is some discussion in the literature review which then briefly mentioned the mechanisms of this. Here a bit more detail would be useful. However, the main point here is that the model construction seems to be centred around creating a framework for capturing the behaviour of phase separating materials such as LFP. As the authors mentioned, silicon is a bit different in that, whilst it does have a significant hysteresis, many comment on this coming about from different silicon phases (amorphous, crystalline etc) which arise during (de)lithiation. This is a bit different to the LFP behaviour and I suppose you want to be clear if you think this framework is applicable for silicon or not.

Thank you for the remark. The structure of the model indeed renders it unsuitable for other lithium-storage mechanisms. We clarified this in the introduction.

- When varying the C-rate there are some interesting behaviours. Perhaps I missed this when reading, however I think the core framework of the model at the higher rates is the same (i.e. capturing these different co-existing phases). In the various works by Bazant et al. they have done extensive work on how this phase separation can be suppressed in LFP at higher rates. Here is an example paper (<https://pubs.acs.org/doi/10.1021/nl202764f>). It would be useful to contrast your work with that of this general phase separation suppression behaviour observed at higher rates leading to a quasi solid solution behaviour.

- The path dependency work is also interesting and somewhat following up on the previous point, it would be useful to see if your modelling framework captures the recently reported effects around OCV recovery after high current pulses shown here (<https://onlinelibrary.wiley.com/doi/full/10.1002/adma.202210937>). Here, this interesting behaviour arises due to this claimed quasi-solid-solution behaviour, which I don't believe your framework includes. This is fine of course, however it would be a useful reflection in the limitations of the modelling approach.

Thank you for bringing this to our attention. As you stated correctly, the model does not incorporate changes in the lithiation mechanism of the material and is therefore solely aiming at simulating the phase transition behavior. We were still able to model the experimental results in the publication by Katrasnik et al. with our model – without any changes in the model structure. We included the discussion under ‘Effects of parameter variation’ and added the simulation

results in the supplementary material. Explaining why the phenomena shown by Katrasnik et al. can be modeled with the PD-ECM, while it does not incorporate changes in the lithiation mechanism, though, would require further work and is not scope of this publication. Therefore, this remains a topic for future research.

- On page 10, lines 279-283, the authors comment on the loss in accuracy at low and high SOC, with this being attributed to residual overpotentials. It would be useful to unpack this a bit more in terms of the origins of this behaviour and this statement, as one might expect the overpotential to be quite low at low currents used to simulate the pseudo-OCV.

Thank you for the remark, we attribute the deviation at extremes SOC to the following effects: Inhomogeneities in the plane or perpendicular directions to the electrode are not taken into account for the simulation of the half cells. Therefore, the degree of lithiation may differ more than considered in the PD-ECM. Additionally, the charge transfer resistance is increasing drastically at extreme SOC. Furthermore, the inhomogeneous lithiation may additionally increase the effects of the increased charge transfer resistance. We added some detail concerning the rise of the internal resistance of the electrode at high SOC to the manuscript.

Minor points

- Define SOC on first use in main text

We now defined all abbreviations used in the abstract again in the main text.

- You mention current and voltage clamps. I understand what you mean here however perhaps this is overly specific and you would be better just to refer to the voltage and current for more physically meaningful text

Thank you for the remark, we changed the wording accordingly.

- When discussing the voltage hysteresis on page 2 line 17-18 it would be useful to give a relative magnitude to give readers a sense of this, especially when you later present modelling accuracy in mV

We added some numbers from literature or own measurements for the listed materials to give a sense of magnitude of the hysteresis contributions.

- On page 2 line 30, the authors mention that the OCV is defined by the charge throughput, whilst this is somewhat true, it might be more accurate to say it is based on an electrodes state of lithiation as charge throughput will be dependent on the capacity of the cell

Thank you for the remark, we changed the sentence accordingly: "To account for the hysteresis of the OCV and thus improve the accuracy of the SOC estimation, a mathematical description of the OCV as a function of the current direction and the stored amount of charge relative to the cell's capacity is required."

- On page 4 line 128 and 131 the authors mention a "horizontal course" and "time course". This could be rephrased to make it more physically meaningful and clear

We changed the phrases to make them more clear and physically meaningful.

The former line 128 is now: “For the charging phase, the voltage remains almost constant after the maximum of the OCPn at 40 % SOC has been exceeded. During the subsequent discharge, the voltage shows a similar behavior, this time though after passing the local minimum of the OCPn at 70 % SOC.”

While the former line 131 is: “For a deeper insight into the characteristics of the model, the course of the OCPn and the SOC within the parallel strings, called SOCn, are shown over time in Fig. 1c) and 1d), respectively.

- In various places throughout the manuscript, the authors comment on “high” and “low” behaviour. It would be useful to quantify this a bit more for greater scientific accuracy

We exchanged some of the expression with quantifications for greater scientific accuracy.

Reviewer #2 (Remarks to the Author):

This paper establishes a probability distributed equivalent circuit model (PD-ECM) motivated by the physical insight into hysteresis, the behavior of which is elaborated and validated with commercial cylindrical cells. The manuscript presents some significant results worthy of publication but should be revised to be more precise and accurate in the wording of the text. I want to put forward the following suggestions to improve the quality of the paper.

(1) The novelty of the work should be explained in detail in the Abstract, Highlights, and Conclusion. The main content should concern your research idea and its experimental effect verification.

We changed the manuscript to emphasize the novelty of our work in the abstract, introduction, and conclusion – while removing these parts from the main content.

(2) The expression of the abstract should be improved. A more detailed presentation of innovation should be conducted as well as the experimental verification.

Thank you for the comment, we changed the abstract to bring more focus on the innovation as well as the experimental verification. Unfortunately, the word limit of the abstract prevents us from going into further details.

(3) More recent literature should be cited and analyzed, such as Improved anti-noise adaptive long short-term memory neural network modeling for the robust remaining useful life prediction of lithium-ion batteries, Improved singular filtering-Gaussian process regression-long short-term memory model for whole-life-cycle remaining capacity estimation of lithium-ion batteries adaptive to fast aging and multi-current variations, and so on.

Thank you for the suggested literature. We added those to the manuscript to emphasize the relevance of hysteresis and impedance modeling to further tasks of battery management.

(4) The expression logic can be improved for your proposed method so that the innovation can

be clearly understood. More expression of the mathematical analysis should be conducted to show your ideas clearly. Also, please try to highlight your proposed method and focus on it.

We improved expression and logic to show our ideas clearly.

(5) Grammar should be checked and improved for the entire content. Please try to make every sentence to be correct and easy to be understood.

We checked the manuscript again and tried to remove all remaining errors. If there are any remaining grammar issues, please ask the editor for language lecturing.

(6) More experimental result comparisons with references should be conducted for advantage discussion.

Thank you for the remark. We added some results from the literature and comparison with our results to the discussion.

(7) Many contents are expressed for the traditional methods for this version. Please pay more attention to your ideas and innovation. Also, please describe your invention briefly for the Conclusion.

We changed the conclusion to focus more on the innovation of our work.

(8) The whole structure should be improved. For this version, there is so much content expressing your experiments and basic principles, but not enough expression for your particular idea and methods.

The structure of the manuscript is defined by the journal. Within this structure we reworked our manuscript to better express the novelty of our ideas.

Reviewer #3 (Remarks to the Author):

itle: A physically motivated voltage hysteresis model using a probability distributed equivalent circuit

In this article, the authors describe a modeling approach to model the voltage hysteresis of first order phase transition materials like LFP. They use a physically motivated ECM to describe not only the voltage hysteresis but also the history-dependency of the rate-capability of LFP electrodes.

As the hysteresis effect in LFP cells is relevant for SOC estimation and understanding LFP-specific phenomena, the paper is on a topic of relevance and general interest to the readers of the journal.

The article is well written with a good flow and most of the important aspects are given.

However, it needs a major revision before being eligible for publication in communications engineering. The following issues should be addressed:

1. As you showed in chapter 3, Kondo et al. already developed a model with consideration of non-monotonic eq. potential curves. As they also do not consider diffusion in solid and liquid phase, this model can easily be transferred into an ECM. So please explain the novelty of your approach more in detail compared to the model of Kondo et al..

As you correctly stated, there are certain similarities between Kondo's and our model. Both aim at describing the behavior of a multi-particle model of a material undergoing a first-order phase transition. The novelty of our approach is the transfer into an ECM, that is capable of real-time calculation of the cell's voltage. The approach is close to application, as the model can be parameterized to a commercial full-cell, with several degrees of freedom to design the distribution of resistors. This allows for an accurate description of even high load cases. Furthermore, the model has been included in a state space representation, making it useable in BMS, e.g., for state estimation.

2. Following on from question 1., in your model description (fig. 6) you assign the same capacity to all particles and you just change the ohmic resistance which is connected in series to the voltage source. Please give more detail why you chose this way and not a distribution of capacity and resistance according to a fitted (gaussian distributed) particle-size distribution (similar approach to Kondo), which would be more physically motivated. Would that add computation-time?

Thank you for asking this question. We would like to emphasize that the initial assumption is that each piece of active material, e.g. capacity, is associated with a specific resistance and that these resistances follow a distribution function. Nevertheless, in the next step, presented as model reduction, "particles" with the same resistance are summarized and represented as a fraction of the capacity associated with a certain resistance. Thus, this is essentially a capacity distribution similar to Kondo's work. This step from distributed resistances to distributed capacity is explained in the section 'Model Reduction'. Note, however, that we are not considering different particle sizes, but different effective resistances connecting the material to the current collector. We added a paragraph to discuss the reason for the model choice in the method section.

3. Please explain more in detail how you parameterized the distribution of resistances. What measurements were used to fit the standard deviation of the distribution? How did you separate these resistances for the current distribution from the series resistance of the cell to not account it twice in the model?

In case of the full-cell simulation the rate capability test was used to choose the proper design parameter of the distribution. We added some details in the method section.

We replaced the ohmic part of the R-2RC model with the distributed resistors of the PD-ECM. We admit that this was not clear in the manuscript. Therefore, we revised the section dealing with the full-cell simulation accordingly.

4. Is the Plett-model simulated for each electrode individually? If not, would the results change (fig. 5)?

The Plett-model aims for simplicity and easy implementation and therefore is simulated for the full-cell as an additional state of the state space model. Since inhomogenities caused by the design of the full-cell would be neglected when separating the half-cells, the results of the Plett model may even get worse. We emphasize that we aimed to apply the Plett method as presented and not to study further modifications of this method.

5. Considering the deviation between the PD-ECM parameterized on half-cell level and the measurements on full-cell level, could those differences arise due to inhomogenities in the full-cell? Have you done full-cell measurements in experimental cells to rule out errors due to inhomogenities which you are not able to attribute with your 2RC-ECM? I am suggesting this, as you see a sloping behavior of the measurement in the plateau regions (more inhomogenities) and smaller errors in the phase transition regions (smaller inhomogenities in the cell).

Thank you for your comment. It is highly probable that these errors are caused by inhomogeneities in the full-cell. Upon closer inspection of the plateaus following the stage transition of graphite, two different slopes are visible for the PD-ECM simulation and the full-cell measurement. This may indicate more inhomogeneous lithiation in the full-cell compared to the half-cell. Due to the parameterization of the PD-ECM using half-cells, these inhomogeneities are not taken into account. Additionally, the model does not account for inhomogeneities in the plane and perpendicular directions of the electrode. We added a brief discussion to the manuscript addressing this issue. Unfortunately, we did not measure experimental full cells since the goal was to create a model that can be used for commercial cells in real-world applications.

REVIEWERS' COMMENTS:

Reviewer #2 (Remarks to the Author):

Thank you to the authors reply to the reviewers comment. These have addressed the points raised and I have no other comments.

Reviewer #3 (Remarks to the Author):

The revised manuscript has fully resolved all the raised questions with clarity and understanding to benefit the readers. It qualifies for the publication.

Reviewer #4 (Remarks to the Author):

Thank you for responding to the given questions and improving the manuscript. My questions were fully answered and I recommend to accept the revised manuscript.

Dear Editor,

as the reviewer's comments below do not address further concerns, there is no response from our side.

Reviewer #2 (Remarks to the Author):

Thank you to the authors reply to the reviewers comment. These have addressed the points raised and I have no other comments.

Reviewer #3 (Remarks to the Author):

The revised manuscript has fully resolved all the raised questions with clarity and understanding to benefit the readers. It qualifies for the publication.

Reviewer #4 (Remarks to the Author):

Thank you for responding to the given questions and improving the manuscript. My questions were fully answered and I recommend to accept the revised manuscript.